# Immunomodulators as Therapeutic Agents in Mitigating the Progression of Parkinson’s Disease

**DOI:** 10.3390/brainsci6040041

**Published:** 2016-09-23

**Authors:** Bethany Grimmig, Josh Morganti, Kevin Nash, Paula C Bickford

**Affiliations:** 1Center of Excellence for Aging and Brain Repair, Department of Neurosurgery and Brain Repair, Morsani College of Medicine, University of South Florida, Tampa, FL 33612, USA; bgrimmig@health.usf.edu; 2Sanders-Brown Center on Aging, Department of Anatomy and Neurobiology, University of Kentucky, Lexington, KY 40508, USA; josh.morganti@uky.edu; 3Byrd Alzheimer’s Institute, Department of Molecular Pharmacology and Physiology, Morsani College of Medicine, University of South Florida, Tampa, FL 33613, USA; Knash@health.usf.edu; 4James A Haley VA Hospital, 13000 Bruce B Downs Blvd, Tampa, FL 33612, USA

**Keywords:** Parkinson’s disease, neuroinflammation, microglia, fractalkine, astaxanthin

## Abstract

Parkinson’s disease (PD) is a common neurodegenerative disorder that primarily afflicts the elderly. It is characterized by motor dysfunction due to extensive neuron loss in the substantia nigra pars compacta. There are multiple biological processes that are negatively impacted during the pathogenesis of PD, and are implicated in the cell death in this region. Neuroinflammation is evidently involved in PD pathology and mitigating the inflammatory cascade has been a therapeutic strategy. Age is the number one risk factor for PD and thus needs to be considered in the context of disease pathology. Here, we discuss the role of neuroinflammation within the context of aging as it applies to the development of PD, and the potential for two representative compounds, fractalkine and astaxanthin, to attenuate the pathophysiology that modulates neurodegeneration that occurs in Parkinson’s disease.

## 1. Parkinson’s Disease

Parkinson’s disease (PD) is a debilitating condition that affects millions of people worldwide. With the development of drugs to treat complications associated with significant morbidity and mortality, patients are living up to 20 years after the diagnosis of PD. Although the use of medications has led to a relative improvement in quality of life, these patients are often substantially disabled, requiring significant health care and compensation for loss of wages. It has been projected that the prevalence of PD will double by 2040, indicating severe economic consequences to come as the incidence increases. There are currently no available medications that prevent or reverse the neurodegeneration that causes these disabilities [1].

Parkinson’s disease is primarily characterized neuroanatomically by the degeneration of the neurons in the substantia nigra pars compacta (SN), resulting in a substantial loss of dopamine (DA) afferents to the striatum and subsequent motor impairment. It is estimated that nearly 50% of dopaminergic cells in the SN have been lost prior to clinical presentation of motor dysfunction. It is now understood that PD pathology extends to extra nigral regions including the locus coeruleus, nucleus basalis of Meynert, peduculopontine nucleus, intralaminar thalamus, and lateral hypothalamus suggesting dysfunction and neurodegeneration in many areas of the brain [2]. It is histopathologically characterized by the formation of Lewy bodies, intraneuronal protein inclusions comprised predominantly of α-synuclein (α-syn). These protein aggregates have been observed throughout the brain, and pathological α-syn deposition is thought to begin in the medulla and spread throughout the midbrain to cortical regions in a manner that corresponds to the onset of clinical symptoms [2]. These protein aggregates are associated with microgliosis [3], and impaired cellular physiology, although the precise mechanisms leading to cytotoxicity are currently unknown. 

The vast majority of Parkinson’s disease cases are classified as idiopathic, but approximately 5% of PD cases are genetically linked. There are several gene mutations that confer susceptibility or are associated with the development of PD, including mutations in leucine-rich repeat kinase 2 (LRRK), PTEN Induced putative kinase 1 (PINK1), and the protein deglycase (DJ1). However, α-syn has proven to have a very strong association and relevance to PD and similar disorders collectively known as synucleinopathies. Abnormalities in the SNCA gene that encodes the α-syn protein are strongly correlated with the development of an autosomal dominant familial form of PD [4]. Multiplications of the SNCA gene are known to increase the expression of the α-syn protein, whereas certain missense mutations of the gene (A53T, A30P, E46K) produce variants of α-syn, both of which have known pathological attributes related to the increased propensity to aggregate. 

The physiological role of α-syn, along with the isoforms β- and γ-synuclein, is related to neurotransmission, and they are primarily located in the synapse. Increased concentrations of α-syn protein or mutated forms of α-syn make the protein susceptible to misfolding and polymerization by self-assembly. These misfolded aggregates have heterogeneous conformations that have not been clearly elucidated. These aggregates are associated with various deleterious biological activities including (1) the permeabilization of the cellular membranes [5], associated with an alteration of intracellular ion concentrations [6]; (2) disruption of energy production through interactions with the mitochondria [7]; and (3) interruption of intracellular transport via physical interference of motor proteins, or direct interaction with organelles and vesicles [8]. The formation of soluble oligomeric species is presumed to be relatively more toxic than fibrils, through the energetically favorable association within the phospholipids that forms pores in the cellular membrane [9]. The amount of LB formations throughout the brain reflects the severity of impairment [10]. However, it has been postulated that the organizing of fibrils into Lewy Bodies may serve as a protective mechanism to divert toxic aggregate species into less harmful formations to preserve normal cellular physiology [11].

## 2. The Interaction with Aging

Aging is a major risk factor for the development of PD [12], evidenced by the fact that the incidence increases for every decade over 50 years of age. While aging contributes to PD, it is also underrepresented in PD research, as it is common for the experimental models to be carried out using young animals. However, the impact of aging is essential to consider in terms of the disease process because many of the physiological changes that occur in aged animals can drive or exacerbate the pathological mechanisms that lead to PD and alter both the course and tempo of the disease. 

The impact of aging on the incidence of PD is compounded by the fact that the dopaminergic neurons of the SN seem to have a unique vulnerability to cellular stressors in the microenvironment [12]. Although it is not completely understood what sets these cells apart from those of other DA pathways, the neurons of this nucleus are more susceptible to the homeostatic disturbances and degenerate significantly more compared to similar or adjacent regions [13]. This can be grossly attributed to high oxidative stress and inflammation. It is known that the SN is exposed to higher levels of oxidation compared to other dopaminergic centers [11]. This is partly due to an unusually high concentration of iron, which through the Fenton reaction, can generate free radicals, as well as nitric oxide released by microglia that densely populate this region [14,15]. In addition to the accumulation of reactive oxygen and nitrogen species, the SN is also ill-equipped to neutralize them, due to a low expression of glutathione, an important endogenous antioxidant molecule, leading to an inevitable increase in oxidative stress [16]. Aging is also associated with increased levels of oxidative stress and altered microglial function and this combined with the SN’s reduced resilience to reactive oxygen species (ROS) could promote neurodegeneration.

The role of neuroinflammation in PD is also particularly important in the context of aging. Inflammation is known to increase with age. This is largely attributed to the age related changes in microglial physiology. Microglia, myeloid derived macrophages, are the resident immune cell of the CNS and constitute 10% of the cell population. Their highly ramified processes are highly motile and constantly survey parenchyma, facilitating the detection and the cellular response to infection and injury. However, microglia are also involved in many homeostatic functions and are now known to be highly active during post-natal synaptic pruning and synaptic plasticity [17,18]. Microglia are known to undergo a phenomenon known as priming with age. This describes a propensity for microglia to attain a pro-inflammatory state and is characterized by 4 principal features: (1) primed microglia will exhibit higher levels of inflammatory mediators and surface makers even in the absence of immune stimulation; (2) microglia also demonstrate hyperactivity upon subsequent activation where they release exaggerated amounts of cytokines and reactive oxygen/nitrogen species; (3) they are also resistant to regulatory mechanisms that typically restore microglia back to their inactivated state, causing them to remain in an aggravated state for a longer period of time [19]; and (4) they do not respond to stimuli, such as IL4, that promote anti-inflammatory factors, angiogenesis, and neurotrophic factor secretion [19,20]. For example, Lee, Ruiz et al. (2013) induced microglial activation in the brains of young and old mice with the application of a cytokine cocktail designed to elicit an M1, pro-inflammatory phenotype (IL1β + IL12) or an anti-inflammatory, M2 phenotype (IL-4 + IL-13). These authors demonstrated that M1 cocktail elicits an increased response microglia from the aged brain, while the same aged cells were less responsive to the M2 cocktail. This phenomenon has been reproduced with isolated microglia and has been termed priming [21].

In order to elucidate the molecular underpinnings of the age-related alterations in microglial function, numerous researchers have begun to assess the differential expression of genes and protein in primary microglia. A recent study using RNAseq comparing the transcriptional profiles of isolated microglia to whole brain has identified the microglial sensome [22]. These authors further assessed the sensome in microglia isolated from aged animals and demonstrated that many of the genes related to sensing endogenous ligands were down regulated, whereas genes pertaining to host defense were up-regulated. Furthermore, another recent study using gene microarrays of isolated microglia identified an up-regulation of NFκB related genes in these aged cells [23]. Sustained microglial activation and microglial priming can perpetuate neurodegeneration by increasing cellular stress both from enhanced release of cytotoxic substances, but also from the loss of trophic support as a result of impaired microglial homeostatic function.

## 3. Role of Neuroinflammation

The precise pathophysiology that precipitates the development of PD is unknown, although it is understood that a few key biological processes are often impacted in patients, including mitochondrial function, proteostaisis, immune function leading to oxidative stress and inflammation. Neuroinflammation is critical factor in the disease process that clearly contributes significantly to the neurodegeneration seen in PD. However, it is difficult to ascertain if inflammation initiates pathological features of PD, or is triggered by the widespread protein aggregation and neuronal death that occurs during disease progression. McGeer et al. (1988) described the presence of increased reactive gliosis and infiltrating T-cells around the SN in post mortem analysis of PD brains [24]. Their observations provided some of the first indications that increased neuroinflammation is associated with DA cell death. Many other studies have reported features of inflammation in post-mortem samples, such as increased levels of inflammatory mediators like iNos and Cox-2 [25], supporting the contribution of increased neuroinflammation during advanced stages of the disease. Similar patterns of activated microglia have been detected in the brain areas associated with clinical symptoms of the disease [26]. Interestingly, this distribution of gliosis was evident in both newly diagnosed patients and those with advanced pathology. Elevated levels of pro-inflammatory cytokines have also been detected in the cerebrospinal fluid and plasma of PD patients at early stages of the disease. These findings suggest that microglial activation may be initiated in early stages and remain an ongoing process throughout the disease. Furthermore, inflammatory insults and injuries are known to increase the risk of developing PD. Traumatic brain injuries (TBI) have been linked to an increased risk of developing the disease in a frequency and severity dependent manner; multiple injuries or injuries requiring hospitalization were more strongly correlated with PD onset [27]. This occurrence is largely attributed to the neuroinflammatory cascade that follows trauma. Certain infections causing neuroinflammation have been known to lead to a post-encephalitic Parkinsonism. This may be because the SN is densely populated by microglia [28], rendering it is especially susceptible to inflammatory stimuli. For example, intracranial injections of lipopolysaccharide (LPS), a bacterial antigen, dramatically activate microglia and leads to nigrostriatal degeneration and motor symptoms of PD [29]. Parkinsonism is a separate condition distinct from PD, but these observations of inflammatory insults leading to cell death may have important implications for PD itself. Taken together, these data suggest a potential role of neuroinflammation in ongoing cell death that occurs in PD.

Furthermore, pathological forms of α-syn are associated with microglial activation in the brains of PD patients and this is consistently recapitulated in animal models. The presence of α-syn aggregates modulates glial activity, often eliciting the release of inflammatory mediators. Codolo et al. (2013) treated monocytes with various species of α-syn to demonstrate that the aggregated and pathogenic forms of the protein can facilitate the secretion of IL-1β though stimulation of the inflammasome [30]. The nitration of α-syn is a common modification associated with pathology thought to be promoted in an oxidative environment. Stimulation of microglial cells with nitrated and aggregated α-syn alters the cellular morphology and transcriptional profile to a pro-inflammatory phenotype, with increased transcription of IL-1β, TNF-α, and IFN-γ as well as induction of NF-Kβ signaling [31]. Additionally, this α-syn activation of microglia seems to be related to phagocytic capacity, as inflammatory cascades and activation of NADPH oxidase are initiated after the glial cells take up the aggregates [3,32,33]. Zhang et al. (2005) demonstrated that impeding phagocytosis of primary microglia attenuates the release of superoxide from these cells when exposed to α-syn in vitro. This detection and engulfment of α-syn seems to be dependent on the FCγ receptor, as FCγ deficient mice are protected the resultant neurodegeneration from AAV driven over expression of α-syn [32,33]. It is thought that this inflammatory response to abnormal α-syn will perpetuate neural dysfunction through the release of cytotoxic compounds that leads to cell death. Many of these studies were done using either glial cell lines or in young animals, neglecting the impact of the pro-inflammatory actions of α-syn within the primed glial environment of an aging brain, thus causing even more damage. In fact, when α-syn is introduced into an aged animal it is more cytotoxic compared to the young controls [34]. 

## 4. Fractalkine as an Anti-Inflammatory Treatment

There is promising pre-clinical experimental evidence to support that reducing inflammation, specifically by suppressing microglial activity, can modify the progression of the loss of DA neurons [35,36,37,38]. Non-steroidal anti-inflammatory drugs have been suggested to reduce the risk of PD onset [39]. Also, the use of minocycline, a tetracycline with pharmaceutical actions that extend beyond the classical antimicrobial activity, has been shown to reduce cell death in a neurotoxic model of PD using 1-methyl-4-phenyl-1,2,3,6-tetrahydropyridine (MPTP). These researchers attributed the neuroprotective effect to the down-regulation of iNOS expression and glial activation [40]. Minocycline was also shown to inhibit apoptosis through a reduction of related inflammatory mediators [41]. Because glial activation is associated with the release of pro-inflammatory factors capable of damaging neurons, this data suggests that mediating the excessive inflammation in PD is a viable therapeutic strategy. There are several signals produced by neurons that have an anti-inflammatory action on microglia, including CD200, CD22, CD47 and fractalkine (FKN, CX3CL1), which could be potential therapeutic targets.

Fractalkine is a protein expressed constitutively by neurons in a membrane bound form that can be cleaved by disintegrin and metalloproteinase (ADAM) 10 and 17. This proteolysis releases a soluble form of the protein, but both isoforms are thought to ligate the cognate receptor, CX3CR1, located on the surface of microglia in the CNS [42]. This neuron-glia interaction serves as an important endogenous mechanism to suppress microglial activation and regulate the output of inflammatory mediators and other damaging molecules [42]. It has been shown that increasing fractalkine levels using an AAV9 gene therapy approach can be neuroprotective [35,36], while the absence of this FKN-CX3CR1 signaling cascade confers susceptibility to neurodegeneration in rodent models of PD. Cardona et al. (2006) demonstrated that CX3CR1 deficiency lead to increase neurotoxicity to both peripheral LPS and MPTP injections. In these experiments, both heterozygous and homozygous mice exhibited increased neurodegeneration and IL-1β expression compared to the intact wild-type controls [43]. There is evidence indicating that fractalkine can regulate microglial function and subsequently reduce inflammation in the CNS. Multiple in vitro studies have established that enhancing fractalkine signaling through application of the ligand or stimulation of the receptor can protect against cell death in culture; FKN-ligand decreases microglial apoptosis and protects against neurotoxicity by both LPS and TNF-α. It has been demonstrated that maintaining or enhancing communication of FKN/CX3CR1 is neuroprotective in multiple rodent models of PD. Pabon et al. (2011) [37] attenuated the neurotoxicity of 6 hydroxydopamine (6OHDA) by delivering a chronic intrastriatal infusion of recombinant fractalkine ligand. This preservation of dopaminergic terminals in the striatum was also associated with decreased microglial activation around the lesion site, indicated by a reduced expression of the MHCII surface marker. To further examine the action of fractalkine in a model of PD where synuclein is introduced in the SN via AAV, Nash et al. (2015) corroborated the neuroprotective effects of FKN by using a viral vector to over express the different isoforms of FKN; membrane-bound (mFKN) or the soluble portion (sFKN). In this study, sFKN reduced DA cell death in young rats with overexpression α-syn in the SN via AAV9. These findings may be due to altered FKN-CX3CR1 signaling [44] and a reduction in pro-inflammatory cytokines and modulation of microglial function into a more protective role (Figure 1). These results suggest that supraphysiological levels of the fractalkine are protective against the neurodegeneration occurring in two separate experimental models of PD. Morganti et al. (2012) [36] also illustrated the importance of neuron-glial communication by administering MPTP to animals deficient in the fractalkine ligand. These FKN knockouts were extremely susceptible to MPTP toxicity, and display both a dramatic loss of TH in the SN and the robust gliosis associated with the lesion site. Not only does this work indicate that FKN-CX3CR1 communication is necessary for moderating cell death and subsequent detrimental inflammatory events, but it also suggests that further enhancing or supplementing this signaling pathway above baseline activity is sufficient to achieve neuroprotection against these potent toxins (Figure 1). However, the action of fractalkine is quite complex; CX3CR1 knockout mice have been shown to increase phagocytosis of amyloid, but decrease phagocytosis of synuclein [44]. It is unclear if this action is mediated by membrane bound fractalkine or the cleaved, soluble form. Several studies have tried to clarify the roles of the differential processing of fractalkine and there does not appear to be a clear conclusion at this point. When an obligate soluble version of fractalkine is used, as described here with a gene therapy approach, it has been shown to be beneficial in models of PD and AD tau pathology [35,36,45]. However, in other models where an obligate soluble fractalkine mouse is used, opposite results have been observed, and this paper concludes that the membrane anchored fractalkine is associated with phagocytosis of Aβ [46].

Much of the current data regarding the therapeutic potential of FKN in the treatment of inflammatory and neurodegenerative conditions stems from studies involving young animals. However, it is essential to note that aged microglia have been shown to be resistant to their regulatory signals [20,47]. Furthermore, communication via the FKN/CX3CR1 axis becomes dysregulated with age and can also contribute to microglial priming and dysfunction. There are age associated changes affecting both the ligand and receptor. It has been has shown that FKN is reduced by in the aged hippocampus [48], although the extent of FKN downregulation in the SN has not yet been fully characterized. Additionally, immune challenges in the aged CNS lead to a prolonged down regulation of the fractalkine receptor that is associated with a sustained inflammatory response, further compromising this protective neuron glia interaction [49]. Therefore, it is imperative to investigate the therapeutic potential of FKN in the treatment of PD in aged animal models; future studies are needed to examine the efficacy of FKN on aged or primed microglia.

## 5. Therapeutic Potential of Astaxanthin

As discussed above, the SN is exposed to high levels of oxidative stress relative to other areas of the brain due to innate features of the neurons that comprise this region [50]. For example, there are low levels of glutathione and high concentration of iron leads to the production of free radicals through the Fenton reaction [51]. Glutathione activity in this region declines with age, further reducing the capacity to manage the accumulation of ROS in the SN. Taken together, these characteristics create an environment of high oxidative stress that can impair neuronal function. 

One natural compound of particular interest is astaxanthin, a naturally occurring xanthophyll carotenoid. It is produced by the marine algae *Heamatococcus Pluvialis* or synthetically derived from carotenoid precursors and used commercially to feed to farmed fish species to increase pigmentation. Astaxanthin has many suggested mechanisms of action that uniquely oppose pathophysiology that underlie Parkinson’s disease including actions as an anti-inflammatory action and improvements in aspects of mitochondrial function, indicating a distinct and promising therapeutic potential in the treatment and management of symptoms in PD patients that are likely more important than its role to simply scavenge free radicals [52,53,54].

Astaxanthin has potent and diverse actions as an antioxidant and is reported to be several times more effective than other carotenoids in its class. The molecular structure of astaxanthin allows it to reduce free radicals in a variety of ways, including absorbing them into the carbon backbone, donating electrons and forming adducts with the reactive species. Although xanthophyll carotenoids are structurally similar, the presence of polar ionone rings at either end of the astaxanthin molecule makes it energetically favorable. The configuration allows the molecule to span across the phospholipid bilayer of cell membranes and protect the membrane against lipid peroxidation [55]. 

There is substantial evidence indicating that treatment with astaxanthin causes a reduction in the markers of cellular stress due to excess ROS production, such as 8-isoprostane, protein carbonyl moieties and 8OHdG [56]. Additionally, astaxanthin has been shown to increase the efficacy of endogenous antioxidant mechanisms in vivo including increasing the expression or activity of glutathione, catalase, thioredoxin reductase and superoxide dismutase (SOD) [57,58]. It has also been shown to upregulate heme-oxygenease 1 (HO-1) through increase in NRF [59,60]. These findings suggest that astaxanthin treatment may help alleviate some of the ongoing oxidative stress that occurs during the progression of PD as it contributes to cellular dysfunction. 

However, in addition to oxidative damage, there are a number of physiological changes that occur with age that exacerbate the cellular stress in the SN. Mitochondrial dynamics, proteosomal efficiency [61] and levels of synuclein [62] are all altered with age, likely rendering the DA cells more vulnerable to neurodegeneration. Aging also leads to microglial priming and may facilitate PD disease progression. Chronic microglial activation results in prolonged exposure to cytotoxic, pro-inflammatory cytokines, increasing cellular stress and ultimately leading to neurodegeneration [33]. Age_induced primed microglia are hyperactive upon subsequent stimulation and release exaggerated amounts of cytokines. They are resistant to reversion back to a state of tissue repair and maintenance of homeostasis, as they are less responsive to regulatory mechanisms [20,63,64]. As stated above, synuclein aggregation may also directly facilitate the release of these inflammatory mediators. It is thought that this inflammatory response to abnormal α-syn will perpetuate neural dysfunction through the release of cytotoxic compounds that overwhelm the DA cells, inevitably leading to cell death. 

Given the range of pathological mechanisms involved in neurodegeneration seen in PD, astaxanthin seems to have a unique potential for the treatment of this disorder. Many diverse biological activities have been described in the literature that are particular relevant to that pathophysiology of PD, as well as normal aging. Based on this knowledge, the interaction of aging and parkinsonian symptoms should be responsive to treatment with astaxanthin. For example, astaxanthin has also been implicated in modulating microglial activity. Experiments using astaxanthin to treat a transformed microglial cell line can reduce the expression of IL-6 and iNOS/NO in vitro when exposed to an immune stimulus such as LPS [65]. These results were corroborated by other studies using aged animals where astaxanthin reduced the release of nitric oxide [56]. These molecules are released in high amounts by activated microglia and are associated with neuronal damage; attenuating the output of inflammatory mediators with astaxanthin may offer some neuroprotection from the inflammatory cascades occurring in the SN.

Some authors have reported alterations in mitochondrial function after astaxanthin treatment. Although most of these studies were conducted in vitro, this is of great interest to the treatment of PD. Mitochondrial dysfunction has been implicated in the etiology of the disorder evidenced by the common toxins that induce Parkinsonism. Both MPTP and rotenone are used to produce Parkinson’s models by selectively targeting mitochondria leading to the death of SN neurons. Multiple genetic mutations of proteins involved in mitochondrial dynamics have been clearly linked to the development of familial Parkinson’s. Furthermore, some of these mitochondrial proteins are associated with a loss of function with age, and may contribute to the increased incidence of diagnosis over the lifespan. 

Furthermore, it has been demonstrated recently that mitochondria are a significant source of oxidative stress not only in these DA neurons, but also in additional nuclei known to degenerate in PD. For example, both the locus coeruleus and SN express L-type calcium channels that allow for an extraneous calcium influx that taxes the mitochondria [66,67]. The presence of these channels and their associated calcium burden has been proposed to be a common physiological feature that contributes to the cellular vulnerability for the brain regions affected in PD. Attenuating this mitochondrial derived oxidative stress that results from calcium overload has led to the use of calcium channel antagonists for the treatment of PD [68]. These dihydropyrdines, specifically israpidine, have been shown to be tolerable and safe among PD patients are now in Phase III clinical trial [69].

The success of these drugs lends support for the therapeutic use of AXT as well. There is substantial evidence to suggest that astaxanthin works at the level of the mitochondria. According to HPLC analysis of cellular fractions, astaxanthin will accumulate in mitochondria, and has the capacity to increase mitochondrial activity as indicated by increased respiration and mitochondrial membrane potential (MMP) [70]. Mitochondrial dysfunction is a common pathophysiological observation in PD, and is recapitulated in the α-synuclein model. It has been shown that treating isolated mitochondria with α-synuclein oligomers induced mitochondrial dysfunction by inhibiting complex 1 and associated with reduced calcium retention time, release of ROS and induced mitochondrial swelling [7]. In specific studies related to PD, astaxanthin has been shown to protect SH-SY5Y cells from 6-OHDA [53]. In a similar experiment, astaxanthin treatment mitigated cytotoxicity in PC12 cells from MPP+ induced cytotoxicity. MPP+ is a toxic metabolite of the dopaminergic neurotoxin MPTP used in experimental animal models of PD [71]. These cell culture results were corroborated by an in vivo study using astaxanthin to prevent the neurodegeneration in the SN in response to dose of MPTP (1 i.p. dose 30 mg/kg daily for 28 days) [54]. This treatment regimen effectively protected against the loss of tyrosine hydroxylase in the SN and striatum after chronic exposure to the neurotoxin. However, it is important to understand that many drugs that have been successful in some pre-clinical models of PD have failed to translate to patients with PD. Developing and testing pre-clinical models involving disease relevant proteins such as α-synuclein and the impact of aging must be considered for future studies.

## 6. Conclusions

Parkinson’s disease is primarily characterized by degeneration of the dopaminergic neurons of the substantia nigra. The pathophysiology underlying this cell death is not yet clearly understood, although it is evident that many biological processes are impaired in this vulnerable brain region, explaining the rapid deterioration of the SN with age. Neuroinflammation is an integral factor perpetuating cellular damage during progression of the disease, and efforts to mitigate the inflammatory cascade have been successful in experimental settings, suggesting that anti-inflammatory treatments are a viable therapeutic strategy to employ in managing Parkinson’s disease. Fractalkine signaling has proven to be a critical pathway in inflammation-mediated cell death that occurs in animal models of PD. Astaxanthin has diverse biological activities that have been reported in the literature, many of which seem to directly oppose the pathological mechanisms involved in neurodegeneration of the SN. Both fractalkine and astaxanthin represent two promising novel therapeutic agents for the treatment and management of PD.

## Figures and Tables

**Figure 1 brainsci-06-00041-f001:**
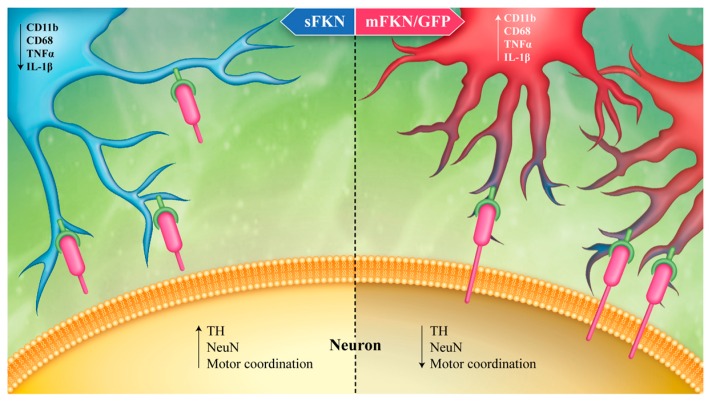
A depiction of ligand/receptor binding for both the soluble and membrane bound isoforms of CX3CL1 (FKN) and their respective influences on the release of inflammatory mediators in the substantia nigra. As discussed above, sFKN when delivered via AAV into CX3CL1-/- mice following MPTP is associated with the suppression of cytokine production. However, when we delivered an obligate membrane bound form of CX3CL1 or a vector with GFP there was no rescue of TH neurons or a reduction in inflammatory mediators.

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
