# Peer review of "Immunomodulators as Therapeutic Agents in Mitigating the Progression of Parkinson’s Disease"

_brainsci, 2016, doi:10.3390/brainsci6040041_

Round 1
Reviewer 1 Report
General comments: The manuscript by Grimmig et al. reviews some of the literature relevant to the proposition that inflammation is a major factor in the etiology of Parkinson’s disease (PD). This is an important topic that merits review. However, this effort has a number of shortcomings that should be corrected. First, it is very focused on dopaminergic neurons in the substantia nigra. But, it is very well documented that the pathology and neurodegneration in PD in not limited to this set of neurons, much less neurons with dopamine as a transmitter. Second, the review of the pros and cons of anti-inflammation therapy is not very thorough and skewed toward one molecule (astaxanthin). Third, the authors seem unaware of the ongoing clinical trials in this arena, which is critical to a review like this. Both isradipine and inosine are in large disease modification trials for PD right now and are not mentioned.
Major concerns:
· The authors need to consider two key facts at the outset of the review. First, the neurodegeneration in PD is not limited to the SNc. There is degeneration that has been documented in a number of other nuclei, including the pedunculopontine nucleus, intralaminar thalamus, lateral hypothalamus and locus ceruleus. Second, the listing of papers exploring the sources of oxidant stress in the SNc are outdated. More recent work has shown that mitochondrial oxidant stress in these neurons is elevated because of their physiological phenotype (see work by Surmeier’s group and others). In fact, the biggest Phase III neuroprotection trial in PD being funded by NIH is based upon this work – and it is not even mentioned! This is a phenotype that they share with other neurons lost in the disease, like locus ceruleus and pedunculopontine neurons (see Sanchez et al.; Kitai et al.).
· What is the evidence that neuroinflammation causes neurodegeneration in idiopathic PD? Starting with the McGeer’s report, there are a number of studies that have shown a correlation, but not causation. The clinical trials evidence on this point is negative. It is also important to distinguish Parkinson’s disease and parkinsonism, which can have a very different pattern of pathology. TBI and encephalitis fall into the parkinsonism category. It is also important to distinguish between models of PD and PD itself. LPS injection can lead to the death of dopaminergic neurons, however, it does not produce a pattern of pathology anything close to that of PD.
· Pg. 4: The statement ‘There is sufficient evidence to support that reducing inflammation, specifically by suppressing microglial activity, is an effective therapeutic strategy’ is just false. Again, the authors have not taken into account the negative clinical data that is out there. At best, this is a hypothesis.
· Pg. 4: The MPTP (and 6-OHDA) toxin models have absolutely no predictive validity in PD. It is an artificial model that produces neurodegeneration in a way that does not mimick that in PD. Every failed clinical trial in PD disease modification has been based upon study of this model and it is high time we stop using it to draw inferences about pathogenesis in PD.
· Pg. 5: There is no evidence that in situ DA generates any signficant level of free radicals. In fact, if this were true, levodopa therapy should accelerate disease progression – it does not.
· Pg. 5: The Shankar paper is cited again here. This paper has major shortcomings and should be carefully considered by the authors before citation (e.g. the use of incorrect statistical approaches, the apparent elevation of antioxidant enzymes in the SN of the young, which decline with age to nominal control levels – quite the opposite of the implications stated).
· The anti-oxidant story. The NIH has spent more money on anti-oxidant therapies in PD than on any other mechanism. All of the trials have failed. Why? What other antioxidants are there in early stages of development? Why might antioxidant therapy be a bad thing? Are there any normal, non-pathological functions of redox signaling that might be disrupted by this approach? (Yes!)
Author Response
see comments to both reviewers attached

Reviewer 2 Report
This review tries to address the timely topic of neuroinflammation in PD and offers two compounds as “novel” therapeutic agents. While the manuscript has some points of interest, it lacks depth and balance, choosing to discuss two agents, fraktalkine pathway and astaxanthin.
Specific suggestions to improve the manuscript:
1. Page 1, line 27, “lethal symptoms” is not quite right. Usually symptoms of PD are not “lethal;” people die of complications.
2. Page 1, paragraph 2, ignores the fact that other brain regions in addition to nigral dopamine neurons are also affected the PD pathology.
3. Page 2, 59-61 the statement is not backed by citations. There are experimental data making this point including in simple cellular systems such published by Tanaka et al (JBC 2004), we well as others that have the opposing view.
4. Page 3, line 116-118 leaves out the well established role of pathogenic genes in PD.
5. Page 3, line 138 brings up post-encephalitic parkinsonism. This is not relevant to PD itself.
6. Page 4, first paragraph, it should be pointed out that minocycline has anti-apoptotic activity as well. Plus, this paragraph does not mention that this agent failed to show an effect in a clinical trial in PD.
7. Page 4, paragraph 2, authors frequently use the word “manipulate” referring to fraktalkine levels as protective. It would be clearer to use a more specific word, such increase or decrease, or enhance or block, to make their points.
8. The figure needs to be cited on page 4.
9. The authors should discuss how a large protein like fraktalkine can be delivered to the brain considering that the pathology in PD is not restricted to the nigra.
10. The authors do not cite the work by Thome et al 2015 about the mechanism of fraktalkine in synuclein mediated neuroinflammation.
11. There is extensive literature about synuclein induced neuroinflammation and its mechanisms that are just mentioned very fleetingly on page 6, end of first paragraph. Expanding this point and discussing the mechanistic role of the agents proposed in this review are recommended.
12. Page 6 last paragraph, it is not surprising that astaxanthin with its antioxidant and mitochondrial effects would protect against MPP+ or 6OHDA in cells or in vivo. There are many other such agents with similar profile, and none have advanced to effective treatments for PD.
13. The manuscript needs minor editing.
14. Author contributions do not reflect the fact that this is a review article and no experiments were involved.
Author Response
See comments to both reviewers in attached file
